# A Five Glutamine-Associated Signature Predicts Prognosis of Prostate Cancer and Links Glutamine Metabolism with Tumor Microenvironment

**DOI:** 10.3390/jcm12062243

**Published:** 2023-03-14

**Authors:** Hai Wang, Yuxiao Chen, Wei Zhao, Haolin Liu, Hongtao Tu, Zhongyou Xia, Rui Wang, Jinze Tang, Chuang Zhu, Rui Li, Xiaodong Liu, Peng Gu

**Affiliations:** 1Department of Urology, The First Affiliated Hospital of Kunming Medical University, Kunming 650032, China; 2The First Affiliated Hospital of Kunming Medical University, Yunnan Province Clinical Research Center for Chronic Kidney Disease, Kunming 650032, China; 3Department of Endocrinology, The First Affiliated Hospital of Kunming Medical University, Kunming 650032, China; 4Department of Urology, Institute of Urology, West China Hospital, Sichuan University, Chengdu 610041, China

**Keywords:** glutamine, prostate cancer, biochemical recurrence, prediction model, tumor microenvironment

## Abstract

Glutamine has been recognized as an important amino acid that provide a variety of intermediate products to fuel biosynthesis. Glutamine metabolism participates in the progression of the tumor via various mechanisms. However, glutamine-metabolism-associated signatures and its significance in prostate cancer are still unclear. In this current study, we identified five genes associated with glutamine metabolism by univariate and Lasso regression analysis and constructed a model to predict the biochemical recurrence free survival (BCRFS) of PCa. Further validation of the prognostic risk model demonstrated a good efficacy in predicting the BCRFS in PCa patients. Interestingly, based on the CIBERSORTx, ssGSEA and ESTIMATE algorithms predictions, we noticed a distinct immune cell infiltration and immune pathway pattern in the prediction of the two risk groups stratified by the risk model. Drug sensitivity prediction revealed that patients in the high-risk group were more suitable for chemotherapy. Last but not least, glutamine deprivation significantly inhibited cell growth in GLUL or ASNS knock down prostate cancer cell lines. Therefore, we proposed a novel prognostic model by using glutamine metabolism genes for PCa patients and identified potential mechanism of PCa progression through glutamine-related tumor microenvironment remodeling.

## 1. Introduction

Prostate cancer (PCa) is the second commonly diagnosed male malignancies and one of the leading causes of male cancer-related death worldwide [1]. Radical prostatectomy is the most effective method of curing localized PCa, and it significantly improves the postoperative survival of patients. However, more than one quarter of patients still experience biochemical recurrence (BCR) after surgery, which subsequently progresses to distant metastasis and PCa-related death [2]. Current predictions of PCa prognosis after curative intent are based on clinical and pathological features whose predictive efficacy remains unsatisfying [3]. Therefore, it is crucial to predict and confirm the BCR status of patients in time in order to classify and manage BCR patients with individualized treatment plans.

As the most abundant and widely used amino acid in human body, glutamine plays an important role in maintaining cellular viability. Recent studies reveal that many types of tumor growth are highly dependent on glutamine, including non-small cell lung cancer, breast cancer and glioblastoma [4]. Likewise, glutamine is also one of the key nutrients that drive PCa progression; patients with higher glutamine levels often have poorer prognoses [5]. A recent study reported that the survival of radiation resistant PCa cells and stem cells relies on a large amount of glutamine, and that the cells undergo growth inhibition after glutamine deprivation [6]. Androgen receptor (AR) can promote the utilization of glutamine in PCa and further improve the survival of PCa cells [7]. Moreover, glutamine metabolism also plays a key role in the dynamics of tumor microenvironment. Glutamine is essential for the immune function of macrophages, lymphocytes, neutrophils and other immune cells [8]. In the tumor microenvironment, both tumor cells and immune infiltrating cells undergo metabolic reprogramming [9]. For instance, tumor-associated macrophages could change the glutamine metabolic state in the tumor microenvironment by secreting IL-23, which promotes the proliferation and activation of regulatory T cells (Tregs) and subsequently inhibits the activity of anti-tumor immune cells [10]. Last but not least, tumor-associated fibroblasts (CAF) are able to compensate for the large demand of glutamine from tumor cells by upregulating the synthesis of glutamine [11]. Because glutamine plays an important function in tumor metabolism, we speculate that glutamine metabolism-based risk models may be helpful to predict and explore the prognosis of prostate cancer. Exploring glutamine metabolism may also help elucidate the mechanism of PCa progression and discover potential therapeutic targets and biomarkers.

In this study, we successfully established a glutamine-metabolism-related model that can predict the risk of BCR in patients with PCa. First, we downloaded transcriptomic data of PRAD from TCGA database to comprehensively analyze the screening of glutamine-related genes and their prognostic value in PRAD. Subsequently, predictive models were constructed and validated in the GEO cohort. We also analyzed the differences in immune infiltration, drug sensitivity and mutation status among patients in different risk groups. Finally, we validated the expression of key genes in clinical samples by qPCR, and in vitro assays were carried out to evaluate the potential biological function of key glutamine-related genes in PCa cells.

## 2. Materials and Methods

### 2.1. Collection of PCa Materials

The training set genes expression profile of 499 PCa samples and 52 controls were downloaded from the TCGA database (https://portal.gdc.cancer.gov, accessed on 1 April 2022). The validation set (GSE70769) data of PCa were obtained from Gene Expression Omnibus database (GEO: https://www.ncbi.nlm.nih.gov/geo/, accessed on 17 May 2022), which was used for the risk model validation. The corresponding clinical information regarding TCGA-PRAD patients was downloaded from the University of California Santa Cruz Xena (UCSC Xena) database (https://xenabrowser.net/datapages/, accessed on 1 April 2022). The corresponding mRNA expression data were analyzed with the “limma” package of R software (version 3.63, creators: Robert Gentleman and Ross Ihaka, city: Auckland, country: New Zealand) to obtain the differentially expressed genes (DEGs). The conditions of screening DEGs were |log2 Fold Change (FC)| > 1 and adjusted (adj.) *p* < 0.05. The glutamine-related genes were then downloaded from the GSEA database (http://www.hmdb.ca, accessed on 1 April 2022):

(GOBP_GLUTAMINE_FAMILY_AMINO_ACID_METABOL TABOLIC_PROCESS;GOBP_GLUTAMINE_FAMILY_AMINO_ACID_CATABOL TABOLIC_PROCESS;HP_ABNORMAL_CIRCULATING_GLUTAMINE_FAMILYAMILY_AMINO_ACID_CONCENTRATION;REACTOME_GLUTAMATE_AND_GLUTAMINE_METABO LTABOLISM;GOBP_GLUTAMINE_FAMILY_AMINO_ACID_BIOSYNT OSYNTHETIC_PRO CESS;HP_ABNORMAL_CIRCULATING_GLUTAMINE_CONCENTRATION).

### 2.2. Construction of Prognostic Model of Glutamine-Related Gene Characteristics

The differentially expressed glutamine-related genes (DEGRGs) in PCa were obtained by the intersection of PCa differential genes and glutamine-related genes. Univariate Cox analysis and lasso regression analysis (R package: “survival” and “glmnet”) were used to screen the glutamine differential genes-related to the prognosis of PCa patients. First, we performed independent prognostic analysis on each gene, screened genes with *p* < 0.05 that were significant for prognosis, then performed lasso regression analysis on these significant differential genes and calculated the risk coefficient (coefi) of each differential gene. Finally, we obtained the risk coefficient of five differential glutamine-related genes. The sum of the expression of each gene in each sample multiplied by the corresponding risk coefficient was used as the risk score of each patient. Then, the median value of the patient’s risk score was taken as the cutoff point, and the patients were divided into high-risk and low-risk groups. The Kaplan–Meier survival analysis was used to compare the biochemical recurrence survival of patients in the high-risk and low-risk groups, and the area under the curve was calculated using the receiver characteristic (ROC) curve (R package, “timeROC”) to evaluate the prediction ability of the model.

### 2.3. Construction of PPI Network

To exploring the functional linkages of each DEGRG with each other, we used the Search Tool for the Retrieval of Interacting Genes (STRING) database (URL: http://string-db.org, accessed on 1 April 2022). Cytoscape software was used to visualize the PPI network.

### 2.4. Validation of External Data Sets

In order to test whether this model has the same prediction ability in other data sets, we verified the risk scoring model by using the GEO data set (GSE70769), which is similar to the above method. The sum of the coefi values of the above five genes multiplied by the expression values of each sample was used as the risk score of the patients, and the patients were divided into high-risk and low-risk groups using the median value as the cutoff point. Kaplan–Meier survival analysis was used to compare the biochemical recurrence survival of patients in the high-risk and low-risk groups. ROC curve was used to calculate the area under the curve to evaluate the prediction ability of the model.

### 2.5. Univariate Cox Analysis and Multivariate Analysis Identified Risk Models

Univariate Cox analysis and multivariate analysis (R package: “survival”) were used to identify whether the risk score could serve as an independent prognostic factor for BCRFS in PCa patients. 

### 2.6. Analysis of the Immune Infiltrate Landscape

To further differentially analyze the molecular characteristics of the scores from the high-risk and low-risk patient groups, immune cell infiltration patterns and immune status were assessed by CIBERSORTx, ssGSEA and ESTIMATE algorithms. The CIBERSORTx database is an online analytical tool that can estimate the abundance of 22 classes of immune cell infiltrates in a sample based on gene expression data. The ssGSEA algorithm from the “GSVA package” was used to estimate immune infiltration score levels in the tumor microenvironment.

### 2.7. Drug Sensitivity Analysis

IC50 (half maximal inhibitory concentration) was an important indicator for evaluating drug efficacy or sample treatment response. We predicted the chemotherapy response of each sample based on the largest publicly available pharmacogenomics database [genomics of drug sensitivity in cancer (GDSC), https://www.cancerrxgene.org/, accessed on 21 April 2022]. The prediction process was implemented by the R package “pRRophetic”.

CellMiner (https://discover.nci.nih.gov/cellminer/home.do, accessed on 21 April 2022) is a database and query tool designed for the cancer research community to facilitate integration and study of molecular and pharmacological data. We applied the CellMiner database to demonstrate whether these risk DEFRGs can predict anticancer drug sensitivity. Only FDA-approved drugs and drugs in clinical trials were included in the analysis. Spearman correlation analysis was performed to determine the correlation between the expression levels of glutamine-related genes and drug sensitivity.

### 2.8. Enrichment Analysis of Pre-Defined Gene Sets Based on the Low and High-Risk Model

The Gene Set Enrichment Analysis (GSEA) was implemented using the GSEA software (Versions:4.2.3). Specifically, we first divided the 422 samples into high-risk and low-risk groups thought the median value of the risk model score and matched the expression matrix of each sample. Then, we uploaded the expression matrix to the GSEA software, chose the Gene Ontology (GO) and Kyoto Encyclopedia of Genes and Genomes (KEGG) gene sets as the indicator gene sets.

### 2.9. Patient Preparation

A total of eight PCa tissues and seven normal benign prostate hypertrophy (BPH) tissues were collected by surgery or needle biopsy from the First Affiliated Hospital of Kunming Medical University. All participants provided written informed consent prior to the study. The experiment was approved by the Institutional Ethics Committee of the First Affiliated Hospital of Kunming Medical University. No PCa patients received any treatment prior to surgery. Finally, the tissues were frozen in liquid nitrogen and then stored in a −80 °C refrigerator pending further experiments.

### 2.10. Total RNA Extraction and Quantitative Real-Time Polymerase Chain Reaction (qRT-PCR)

The total RNA of eight PCa and seven prostate normal tissue samples were lysed with TRIzol reagent (yanshunbio, Shenzhen, China) and isolated following the manufacturer’s instructions. The concentration and purity of the total RNA solution were quantified using the NanoDrop 5000 spectrometer (BioTeKe Corporation, Beijing, China). Before qRT-PCR, the isolated RNA was reverse-transcribed to cDNA. A total volume of 20 uL reaction system was reached using 1000 ng of total RNA: 4 uL 5× PrimeScript RT Master MIX; total RNA volume was calculated based on the concentration; and finally, 20 uL of reaction system was reached by adding enzyme-free water. After gentle mixing, the reverse transcription reaction was carried out under the following reaction conditions: 37 °C for 15 min, 85 °C for 5 s, and finally stored at 4 °C. The qRT-PCR reaction mixtures consisted of 2ul of cDNA solution, 10 uL TB Green Premix Ex Taq II (Tli RNaseH Plus) (2×), 0.8 uLeach of forward and reverse primer, 0.4 uL ROX Reference Dye II (50×) and 6 ul RAase Free dH2O. The PCR reaction process was performed under the following conditions: 40 cycles that each involved incubation at 95 °C for 65 s and 60 °C for 94 s. The forward and reverse primers of 5 key genes and GAPDH were shown in Appendix A. All primers were synthesized by Servicebio (Servicebio, Wuhan, China). The expressions of these genes were normalized by the expression of GAPDH, and the relative expression of 5 key mRNAs was determined using the 2^−ΔΔCt^ method.

### 2.11. Cell Culture

DU145 and 22Rv1 human prostate cancer cell lines were obtained from Kunming Animal Research Institute (Kunming, China). DU145 and 22Rv1 cells were cultured in RPMI-1640 (Gibco, New York, NY, USA) medium with 10% fetal bovine serum (FBS, BI, Haemek, Israel), 100 U/mL penicillin and streptomycin. The cells were grown in an incubator with 5% CO_2_ at 37 °C.

### 2.12. siRNA Transfection

Small interfering RNA (siRNA) duplexes targeting the human GLUL and ASNS genes were synthesized and purified by GenePharma (Suzhou, China). Appendix A shows siRNA sequences. DU145 or 22Rv1 cells (2 × 10^5^~3 × 10^5^) were seeded in 6-well plates, and siRNA was mixed with Lipofectamine-3000 (Invitrogen, Waltham, MA, USA) for transient transfection, according to the manufacturer’s instructions.

### 2.13. Cell Proliferation Assays

Cell proliferation was determined by CCK8 reagents. The cells transfected with the control siRNA or ASNS siRNA (si-ASNS) and GLUL siRNA (si-GLUL) were trypsinized from the 6-well plates at 24 h after transfection, and 2000 cells in a volume of 100 uL complete medium or medium without glutamine were seeded into each well of 96-well plate, repeated in three holes. One plate of cells was cultured with complete medium, and the other one was cultured with no glutamine medium. The first day of seeding was treated as the 0 h time point. CCK8 assays were performed at 24 h intervals, and 10 μL of CCK8 reagents were added to each well and incubated for 1.5 h under normal cell culture conditions. Absorbance was measured at 450 nm with a microplate reader.

### 2.14. Statistical Analysis

All data analyses were performed using R software (version 3.6.3), and appropriate R packages were selected. *p* < 0.05 was considered to be a significant statistical difference. * *p* < 0.05, ** *p* < 0.01, *** *p* < 0.001, **** *p* < 0.0001, ns represents no significant difference.

## 3. Result:

### 3.1. Screening and Identification of Genes Related to Differential Glutamine Metabolism in PCa

First, samples with incomplete clinical information were excluded from further analysis, and the clinical characteristics of the 422 PCa samples are listed in Table 1. A total of 5926 DGEs were recognized between PCa and prostate normal tissue after differential expression analysis by R package “limma”. To identify genes related to glutamine metabolism, 91 glutamine-related genes were gathered from the GSEA database (Appendix A). Then, 40 DEGRGs of PCa were obtained by taking the intersection of TCGA DEGs and glutamine-related genes (Figure 1A). The correlation heatmap showed the expression difference of 40 glutamine-related DEGs between prostate cancer tissue and normal prostate tissue (Figure 1B). Finally, we determined the protein interaction among the 40 glutamine-related DEG through PPI network analysis. (Figure 1C).

### 3.2. Screening Glutamine Metabolism-Related Genes and Construction of Risk Scoring Model

First of all, we found nine genes that were significantly related to BCR-free survival (BCRFS) by univariate Cox analysis (Figure 2A). The five key genes (ATCAY, GLUL, ASNS, CAD, FPGS) and the correlation coefficient were identified by Lasso regression (Tenfold cross-validation method) (Figure 2B,C). The correlation analysis revealed that all five genes were independent risk factors of PCa (Figure 2D,E). Then, based on the correlation coefficient and expression values of the five genes (Table 2), a prognostic model for predicting biochemical recurrence in each patient was constructed as follows: risk score = (−0.204340092 ∗ ATCAY) + (−0.130197387 ∗ GLUL) + (0.275033455 ∗ ASNS) + (0.63782178 ∗ CAD) + (0.679691817 ∗ FPGS). Next, PCa patients in the TCGA cohort were divided into two groups (high-risk and low-risk) according to the median value of the risk model score. As a result, we found that the proportion of PCa patients with biochemical recurrence were significantly higher in the high-risk group than those in the low-risk group. Consistent with our findings, we also found that ATCAY and GLUL were highly expressed in the low-risk group, while ASNS, CAD and FPGS were upregulated in the high-risk group (Figure 2F). Subsequently, a Kaplan–Meier survival curve showed that BCRFS of patients in high-risk group was significantly shorter than those in the low-risk group (*p* < 0.05) (Figure 2G). Additionally, the areas under the 1-year and 3-year time-dependent ROC curves were 0.697 and 0.72, indicating the good effectiveness of the prognostic risk score model at the 1-year and 3-year points. However, the AUC of the 5-year time-dependent ROC curve was 0.642, showing a poor effectiveness of the prognostic risk score model at the 5-year point (Figure 2H).

### 3.3. Clinical Characteristics and Survival Analysis in Different Risk Groups

We conducted a correlation analysis between clinical characteristics and biochemical-recurrence-free survival in PCa patients from TCGA cohort. We found that patients with Gleason score > 7, N1, stage ≥ T3 had shorter biochemical-recurrence-free survival (Figure 3A). Additionally, the risk score was significantly higher in patients with biochemical recurrence, Gleason score > 7, N1 and stage ≥ T3 (*p* < 0.05), (Figure 3B). Moreover, we also demonstrated that the predictive value of the risk score was more efficient in PCa patients with Gleason score > 7, N0 stage, stage ≥ T3 and age < 65, while the BCRFS was not statistically different among PCa patients with Gleason score < 7, N1 stage, T2 stage and age < 65 (Figure 3C–F).

### 3.4. Verification of Risk Scoring Model by External Cohorts

Since we evaluated the predictive efficacy of the risk model by using PRAD cohort in TCGA database, we further verified the predictive power of this model by downloading the PCa dataset (GSE70769) from the GEO database. Similarly, we divided all patients into high-risk and low-risk groups using the median value and according to the risk score of each patient. Similar to the TCGA training set, a higher proportion of PCa patients with biochemical recurrence were observed in the high-risk group than in the low-risk group. Moreover, the expression pattern of ATCAY, GLUL, ASNS, CAD and FPGS in the high-risk and low-risk groups was the same as the training set (Figure 4A). Similarly, Kaplan–Meier survival analysis showed that BCRFS of patients in the high-risk group was significantly shorter than those in the low-risk group (*p* < 0.05) (Figure 4B). The areas under the 1-year, 3-year and 5-year time-dependent ROC curves were 0.643, 0.699 and 0.690, respectively (Figure 4C). Combined with the above research results, this model is shown to have good prediction ability.

### 3.5. The Glutamine-Related Risk Score Was an Independent Predictor of Biochemical Recurrence of PCa

First, we applied univariate analysis and found that the risk score, Gleason score, PSA value and pathological T and N stages were significantly associated with biochemical recurrence of PCa (*p* < 0.05, Figure 5A). Next, multivariate Cox regression analysis found that both the Gleason score and the risk score were independent prognostic factors for biochemical recurrence of PCa (*p* < 0.05, Figure 5B). These results indicate that the risk score model constructed with the 5 glutamine-related genes was an independent risk factor for the BCR of PCa patients.

### 3.6. Immunosuppressive Tumor Microenvironment and Pathways Were Enriched in High Risk Group

We then analyzed the differential infiltrated immune cell types by CIBERSORTx and identified that naive B cells, resting mast cells, and neutrophils were more enriched in the low-risk group, whereas Tregs, M2 macrophages, and activated NK cells were significantly enriched in the high-risk group (Figure 6A). We also calculated the stromal score, estimate score, and immune score by the ESTIMATE algorithm and discovered that the scores were significantly lower in the high-risk compared with the low-risk group, whereas tumor purity was significantly higher in the high-risk compared with the low-risk group (Figure 6B). We subsequently evaluated the association between immune activity and risk scores by using the single sample gene set enrichment analysis (ssGSEA), Interestingly, we also observed that patients with high risk scores had a lower of type II IFN response scores than those with low risk scores (Figure 6C). Finally, we noticed that the proportion of M2 macrophages was positively correlated with Gleason score, ASNS, CAD, FPGS and risk score. Similarly, the Tregs fraction was also positively correlated with Gleason score, negatively correlated with ATCAY (*p* < 0.05), and not significantly correlated with risk score (*p* > 0.05) (Figure 6D). Collectively, these results show that patients with high glutamine metabolic signatures might receive immunosuppressive effects through various mechanisms. The results also indicated that abnormal glutamine metabolism may contribute to the malignant progression of PCa through modification of tumor microenvironment, especially through altering the infiltration of immune-related cells and pathways.

### 3.7. Mutational Landscape and the Potential Molecular Mechanisms in High- and Low-Risk Groups of PCa

We analyzed tumor mutational burden (TMB) levels in patients from different risk groups and found that patients in the high-risk group had significantly higher TMB levels than those from the low-risk group, and the risk score was positively associated with TMB (Figure 7A). We also determined the level of microsatellite instability (MSI) scores in high-risk and low-risk groups, and, similar to TMB, MSI scores in the high-risk group were significantly higher than those in low-risk group, and the risk score was positively associated with MSI (Figure 7B). Subsequently, the top 20 most frequently mutated genes were analyzed in high-risk-group patients, and 148 out of 211 patients with mutations were detected in the high-risk group, and the three genes with the highest mutation rates were TP53 (22.3%), TTN (20.3%) and SPOP (12.8%) (Figure 7C). By contrast, in the low-risk group, 116 out of 211 patients harbored genetic mutations, including SPOP (23.3%), TP53 (14.7%) and TTN (12.9%) (Figure 7D). It is worth noting that TP53 is a typical tumor suppressor, and its mutation is one of the common factors in tumorigenesis. TP53 mutations are even more directly associated with advanced prostate cancer and enhance the aggressiveness of prostate cancer [12,13]. Moreover, SPOP mutation is a common mutation pattern in prostate cancer, accounting for about 15% of primary prostate cancers [14]. Interestingly, the frequency of SPOP mutations was significantly higher in primary prostate cancers than in metastatic prostate cancers [15]. This is similar to our results that patients in the low-risk group had a higher frequency of SPOP mutations. Furthermore, single-sample GSEA was used to analyze the potential molecular mechanisms between different risk groups. Interestingly, pathways associated with the repair of DNA damage were enriched in the high-risk group, for example, “Base excision repair (BER)”, “Nucleotide excision repair (NER)”, “Homologous recombination (HR)” and “DNA mismatch repair (MMR)” (Figure 7E–H). Other pathways were mainly related to metabolism, like “Pyrimidine metabolism”, “Glyoxylate and dicarboxylate metabolism” and “Oxidative phosphorylation”, etc. (Appendix A).

### 3.8. Drug Sensitivity Analysis

In addition to surgical treatment modalities, endocrine therapy and chemotherapy are both systemic treatment for patients with PCa. We calculated individual drug half maximal inhibitory concentrations (IC50) for each sample in high-risk and low-risk groups using the R package “pRRophetic”. We found that the IC50 of the antiandrogen bicalutamide was significantly lower in the low-risk group than in the high-risk group (Figure 8A). In addition, we noted that patients in the high-risk group appeared to derive better survival benefit from chemotherapeutic agents, such as docetaxel, doxorubicin, etoposide and mitomycin C (Figure 8B–E). Next, we evaluated the correlation between the drugs in the CellMiner database and the five DEGRGs by Spearman correlation analysis, and 16 drugs with the greatest correlation with these 5 DEGRGs were retained (Figure 8F). Collectively, our data demonstrated that the glutamine-related risk signature could be applied as a useful clinical parameter to choose proper treatment for PCa patients.

### 3.9. Validation of 5 Key Genes Expression Levels in PCa Clinical Samples

We validated the expression levels of 5 prognostic genes in the TCGA-PRAD (Appendix A) and found that both ATCAY and GLUL were lower expressed in the tumor tissues than normal prostate tissues, while the ASNS, CAD and FPGS were overexpressed in the prostate cancer group. We further validated the expression of these 5 genes in clinical samples of PCa (*n* = 8) and benign prostatic hypertrophy (*n* = 7) by qPCR. We revealed that the expression of ATCAY and GLUL were significantly downregulated, whereas CAD and FPGS were upregulated in PCa samples. The results were consistent with our bioinformatics results (Figure 9A,B,D,E). However, the expression of ASNS was not statistically different between PCa and normal samples (Figure 9C). This outcome may result from the insufficient sample size. 

### 3.10. Silencing of ASNS and GLUL in PCa Cell Lines Induced Tumor Growth Inhibition under Glutamine Ablation

As ASNS and GLUL are abundantly expressed in prostate tissue, we sought to determine their biological function through PCa cells. We knocked down the expression of ASNS and GLUL in DU145 and 22Rv1 cell lines by siRNA. To confirm the knockdown efficacy, qPCR assays were conducted (Figure 10A,B). Furthermore, cell growth was evaluated by CCK8 proliferation assay. We observed that knocking down either ASNS or GLUL did not lead to a significantly different proliferation both in the DU145 and 22Rv1 cell lines (Figure 10C,D). Surprisingly, we also found that, under the glutamine deprived condition, PCa cells displayed significant growth inhibition after ASNS or GLUL knocking down (Figure 10E,F). Our results suggest that PCa cells are highly dependent on both exogenous and endogenous glutamine, and that targeting glutamine metabolism may be effective in inhibiting PCa growth.

## 4. Discussion

Metabolic reprogramming is considered to be one important hallmark of tumorigenesis and tumor development [16]. Like metabolization in normal cells, tumor cells depend on the Warburg effect and high glutamine metabolic state to meet their high energy requirements [17]. Glutamine plays a critical role in multiple biological activities, including tumor growth and metastasis. For example, glutamine participates in the tricarboxylic acid cycle (TCA) as an intermediator, helps to construct purine and pyrimidine bases, and synthesizes glutathione to promote the antioxidant capacity of cancer cells, etc. [18]. More importantly, glutamine not only influences the development of tumor cells, it also regulates the metabolic state throughout the tumor microenvironment. In the TME, tumor cells take up the maximum amount of glutamine to meet growth consumption, resulting in lack of glutamine utilization by immune cells and reduced anti-tumor immune response [19]. Moreover, glutamine metabolites can also promote tumor progression by inhibiting glucose-dependent differentiation of macrophages as well as T cells, therefore disrupting the function of immune cells [20]. Therefore, targeting glutamine metabolism has been proposed as a potential approach for cancer treatment.

The occurrence of biochemical recurrence in patients with PCa has been an intractable clinical problem. Biochemical recurrence occurs in approximately 20–40% of patients after radical prostatectomy or radiotherapy and is associated with poor clinical outcomes [21]. Although a variety of treatment options are available to improve the prognostic outcome of PCa patients [22], patients with biochemical recurrence were likely to develop distant metastasis, which eventually led to cancer-related death [3]. In clinical practice, surgical margins, Gleason Score, PSA and seminal vesicle invasion are useful prognostic factors of BCR in PCa patients [23]. Therefore, accurate prediction of BCR is of great importance for long-term survival of PCa patients. In this study, we identified a glutamine metabolic signature consist of 5 genes that could predict the prognosis, especially BCRFS, for PCa patients. The univariate and multivariate Cox regression analysis also showed that the signature was an independent risk factor of BCRFS in PCa. However, multivariate Cox regression analysis showed that the HR for the prediction model-generated risk score is relatively low (close to 1), which could be a flaw in this study. Further training and validation of this model with larger sample size may be a solution. In the present study, the model demonstrated a good efficacy and accuracy in both TCGA and GEO cohorts. As a result, early intervention and individualized treatment strategies may be developed according to the risk stratification. Currently, BCRFS prediction models based on other characteristics have also been established. For example, the BCRFS model of PCa was constructed through differential glycolysis-related gene characteristics [24]. An iron-death-related prognostic model for BCRFS was also reported, and differences in immune infiltration in tumor microenvironment under different molecular clusters was observed [25]. In addition, a recent study reported a long, non-coding RNA-based molecular feature model to predict BCRFS [26]. Compared with these molecular classifiers, our glutamine-metabolism-based genetic signature demonstrated comparable predictive efficacy in predicting the BCRFS in PCa patients. Collectively, this glutamine-metabolism-related risk signature may serve as a useful parameter in stratifying patients during clinical decision making.

In this present study, the prognostic model was constructed with five genetic features: Kinesin Light Chain Interacting Caytaxin (ATCAY), Glutamate-Ammonia Ligase or Glutamine Synthase (GLUL), Asparagine Synthetase (ASNS), Carbamoyl-Phosphate Synthetase2, Aspartate Transcarbamylase, And Dihydroorotase (CAD) and Folylpolyglutamate Synthase (FPGS). Some of these genes are important regulators during cancer progression. For instance, GLUL catalyzes the synthesis of glutamine from glutamic acid and ammonia and plays an important role in maintaining acid–base homeostasis, cell signaling and growth, ammonia detoxification and pathological angiogenesis [27,28]. Its overexpression can be used as an early marker of hepatocellular carcinoma [29]. Enhanced expression of GLUL is also related to the malignant progression of pancreatic cancer, so targeting GLUL can effectively reduce the survival of pancreatic cancer cells [30]. Moreover, ASNS is highly expressed in PCa, especially in castration resistant prostate cancer (CRPC) patients, and inhibition of ASNS can lead to decreased viability of PCa cells. As a result, targeting ASNS was proposed as an potential strategy to treat CRPC [31]. Importantly, CAD upregulation in PCa promotes androgen receptor (AR) nuclear translocation and transcriptional activity, therefore enhancing the metastatic capacity and recurrence risk of PCa [32]. In our current model, ATCAY and GLUL are highly expressed in low-risk patients and patients without biochemical recurrence. On the contrary, the expression of ASNS, CAD and FPGS were upregulated in high-risk patients and patients with biochemical recurrence. Combined with the previous data, we believed that ATCAY and GLUL may be a protective factor in PCa, while ASNS, CAD and FPGS are act as unfavorable factors to promote the occurrence and development of PCa. Unfortunately, ATCAY, GLUL and FPGS have not been studied in depth in PCa, so evidence of their mechanisms remains limited.

Tumor microenvironment (TME) is one of the most important factors that promotes malignant progression of solid tumors. A large number of immune cells and cytokines in the microenvironment constitute an immunosuppressive environment that promotes tumorigenesis, metastasis, and therapeutic resistance of malignancies [33]. Normally, M2 macrophages has been considered as tumor-associated macrophages (TAM) that support the development of numerous tumors [34]. As an important immune component of the prostate cancer tumor microenvironment, the total amount of TAM infiltration and its subtype are closely related to the clinical outcome and pathological features of prostate cancer patients, and prostate cancer patients with larger amount of infiltrating TAM often have a worse prognosis [35,36,37]. Tregs can inhibit anti-tumor immune responses and promote the formation of an immunosuppressive microenvironment, which, in turn, promotes cancer progression. Tregs tend to be highly infiltrative in tumor tissues, including hepatocellular carcinoma, lung cancer, pancreatic cancer, breast cancer and prostate cancer, and are directly related to poor prognosis [38,39]. It is reported that Tregs can modulate the direct development of post atrophic hyperplasia (PAH) into prostate cancer and suppress the anti-tumor immune response before the primary tumor is formed [40]. In addition, glutamine metabolism can also participate in the remolding of TME through multiple mechanisms. For example, tumor cells can compete with immune cells for glutamine uptake and reduce the viability of immune cells to promote tumor progression. Glutamine also alters the metabolism of immune cells in the tumor microenvironment and regulates the differentiation and function of immune cells in many ways. For instance, M2 macrophages consume more glutamine intracellularly than M1 macrophages, and glutamine also promotes macrophage polarization to M2 type by modifying the correspondent gene expression [19]. Additionally, PCa cells can alter the phenotype of recruited macrophages by releasing chemokines to polarize them toward a pro-tumor macrophage mixed state [41]. Notably, in the current study, we found a significantly higher infiltration of M2 type macrophages and Tregs cells in the high-risk group. Available data suggest that Tregs cells and M2 macrophages are important regulators in driving the malignant progression of PCa and are associated with increased risk of biochemical recurrence of PCa [37]. Finally, we also found that patients in the high-risk group had higher tumor purity scores, but lower stromal, immune and estimate scores, indicating a more intense immunosuppressive effect. As a result, we proposed a potential mechanism that aberrant glutamine metabolism may contribute to progression of PCa via modifying TME, especially immune cell infiltration. However, future mechanistic study is warranted.

Pembrolizumab, the anti–programmed cell death protein 1 (PD-1) antibody, is approved by the FDA for the treatment of microsatellite instability–high (MSI-H) and TMB-high (TMB-H) tumors [42,43]. Available research data indicate that patients with TMB-H and MSI-H scores may benefit from immunotherapy [44,45]. A growing number of studies have shown that TMB can be used as a predictive marker for response to immune checkpoint inhibitors (ICIs) in a variety of tumors, including non-small cell lung cancer and melanoma [43]. Deficient mismatch repair (dMMR)/microsatellite instability (MSI) usually occurs in colorectal cancer and can be detected in about 15% of colorectal cancer patients [46]. Although MSI is uncommon in prostate cancer, with an incidence of about 3.1%, it is important in guiding clinical treatment of prostate cancer [47]. In our study, many pathways related to the repair of DNA damage were enriched in the high-risk group by GSEA analysis, for example, “Base excision repair (BER)”, “Nucleotide excision repair (NER)”, “Homologous recombination (HR)” and “DNA mismatch repair (MMR)”, which may result in mutations in high-risk group patients. We observed that the patients with high risk scores tended to have higher TMB and MSI scores, which indicated that patients with high risk scores would be more sensitive to immunotherapy. Additionally, we observed that patients in the high-risk group appeared to derive better survival benefits from chemotherapeutic agents such as docetaxel, doxorubicin, etoposide and mitomycin C. Therefore, immunotherapy plus chemotherapy may be a new option for PCa patients with high glutamine profile.

Although our study achieved some interesting results, there are still some limitations. First, there was a small amount of data for PCa patients; although the TCGA database is a large public database, there were only 499 prostate cancer patients, and only 422 cases were carefully screened by us with information on biochemical recurrence. The number of prostate cancer patients in the GSE70769 dataset was also low, and the clinical information of some patients with PCa from the GSE70769 dataset was incomplete. Second, this was a retrospective analysis, and selection inaccuracies may exist in this study. Third, although we initially demonstrated the important role of glutamine in prostate cancer cells through qPCR and siRNA silencing of target genes in this experiment, extensive in vivo and in vitro experiments are still needed to further explore the potential mechanisms behind the regulation of glutamine metabolism in PCa risk score and BCR.

## 5. Conclusions

Our study identified the key glutamine-metabolism-related genes that described the glutamine-metabolism background in PCa. We constructed a risk signature with high accuracy in predicting BCRFS and treatment response in PCa patients. Genetic mutation, the landscape of TME and drug sensitivity were compared according to our risk stratification. Moreover, the study provided new insights into the mechanisms by which altered glutamine metabolism regulates the progression of PCa by modulating the dynamics of TME. Further relevant in vitro and in vivo verification are warranted to deeply explore the mechanisms involved in the metabolic regulation of glutamine synthesis in PCa.

## Figures and Tables

**Figure 1 jcm-12-02243-f001:**
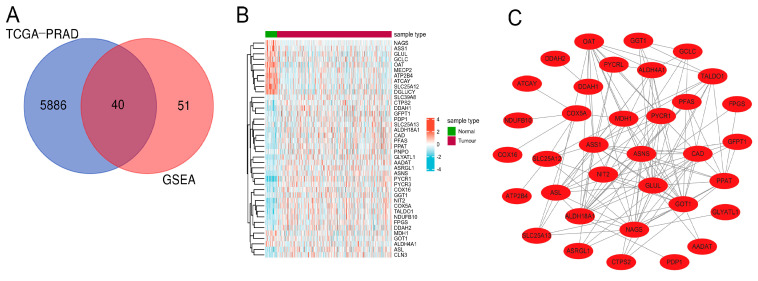
Preliminary screening and identification of PRAD-related DEGRGs. (**A**) The 40 DEGRGs overlapping from the TCGA and GSEA database are shown by Venn diagram. (**B**) The correlation Heatmap shows the expression difference of 40 DEGRGs. (**C**) PPI network of 40 DEGRGs.

**Figure 2 jcm-12-02243-f002:**
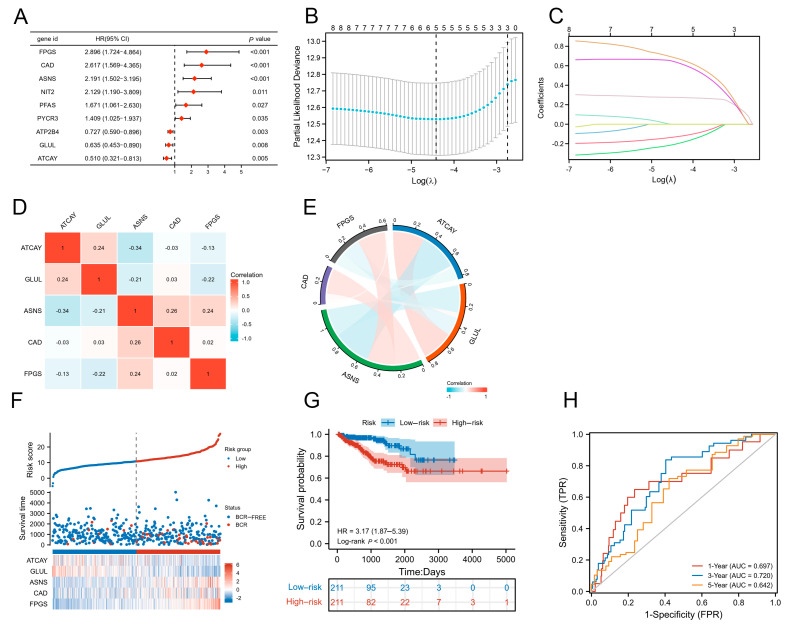
Screening DEGRGs related to prognosis of prostate cancer. (**A**) Univariate Cox regression analysis of glutamine-associated genes. (**B**,**C**) Screening glutamine-associated genes with patients’ prognosis by lasso regression analysis. (**C**) represents the variables with non-zero coefficients. From top to bottom, the first line (yellow) represents FPGS, the second line (pale purple) represents CAD, the third line (pale gray) represents ASNS, the fourth line (pale blue) represents PFAS, the fifth line (pale yellow) represents NIT2, the sixth line (pale green) represents PYCR3, the seventh line (pale red) represents GLUL, and the eighth line (blue) represents ATCAY. To ensure the accuracy of the model, lambda.min (0.0120) was chosen as λ to construct the model and screen out five genes, including ATCAY, GLUL, CAD, ASNS, and FPGS. (**D**,**E**) Independence analysis between the five genes. (**F**) The distribution of the five-gene risk score, the five-gene expression differences in high and low risk groups, and survival status for each patient. (**G**) Kaplan–Meier curve of BCRFS in high-risk and low-risk group. (**H**) The 1-year, 3-year and 5-year time-dependent ROC curves of the glutamine-related signature for prediction of BCRFS of patients with PCa.

**Figure 3 jcm-12-02243-f003:**
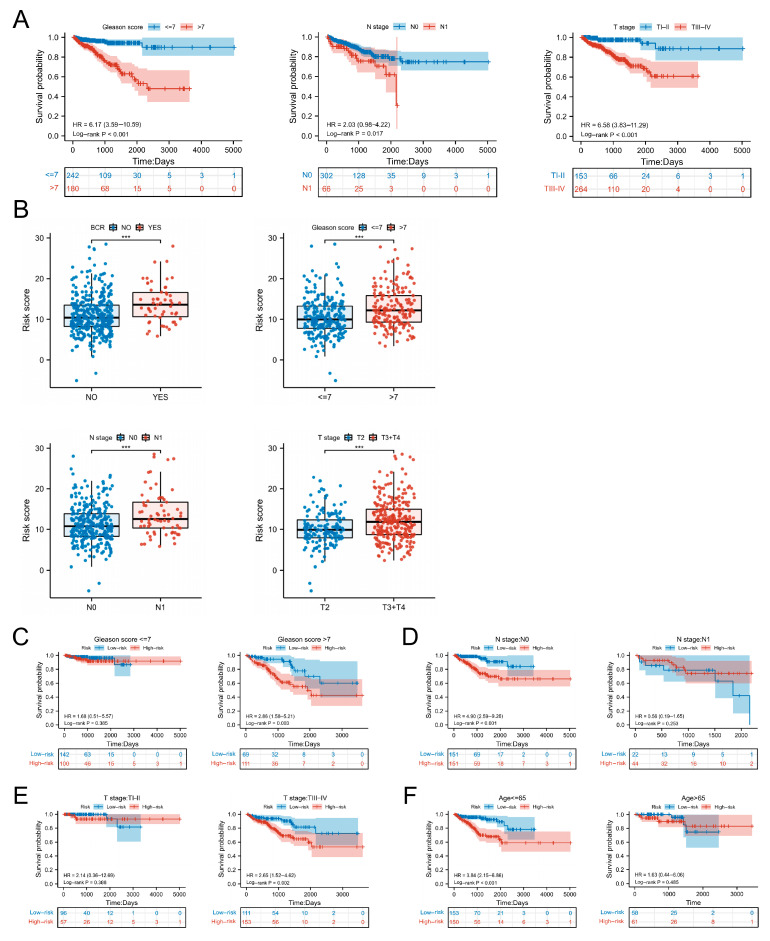
BCRFS analysis in different clinical characteristics. (**A**) BCRFS analysis in different pathological conditions. (**B**) Risk score difference in different pathological conditions. (**C**) BCRFS analysis in Gleason score > 7 and Gleason score ≤ 7. (**D**) BCRFS analysis in N0 stage and N1 stage. (**E**) BCRFS analysis in TI-II stage and TIII-IV stage. (**F**) BCRFS analysis in age > 65 and age ≤ 65. *** *p* < 0.001.

**Figure 4 jcm-12-02243-f004:**
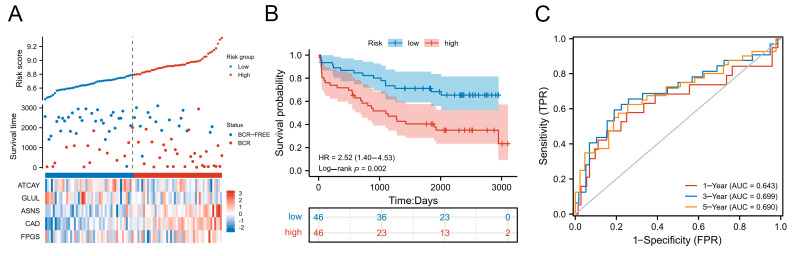
Verification of risk scoring model by external cohorts derived from the GEO dataset (GSE70769). (**A**) The distribution of the five-gene risk score, the five-gene expression differences in high and low risk groups, and survival status for each patient. (**B**) Kaplan–Meier curve of BCRFS in high-risk and low-risk group. (**C**) The 1-year, 3-year and 5-year time-dependent ROC curves of BCRFS in validation datasets.

**Figure 5 jcm-12-02243-f005:**
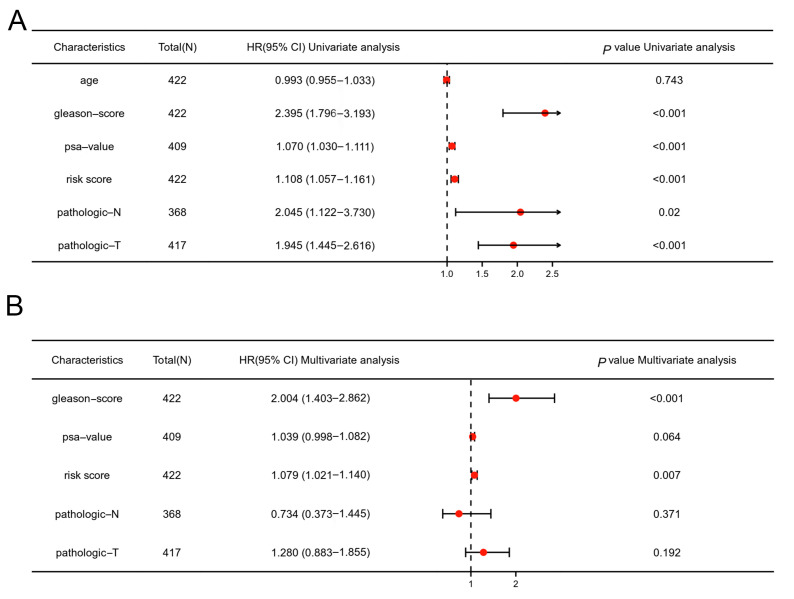
The univariate and multivariate Cox regression analysis of clinical characteristics and risk score. (**A**) Univariate analysis of age, Gleason score, PSA value, risk score and pathological T and N stages. (**B**) Multivariate Cox regression analysis of Gleason score, PSA value, risk score and pathological T and N stages.

**Figure 6 jcm-12-02243-f006:**
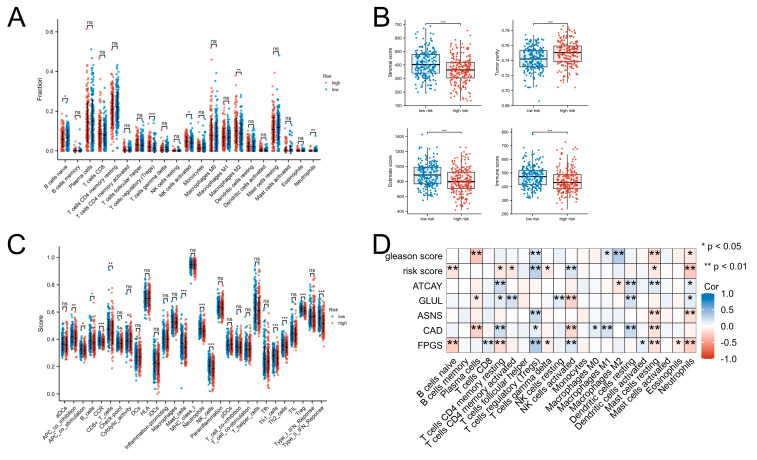
Immune infiltration characteristics in the different risk groups. (**A**) The immune infiltration estimations of the content of 22 immune cells using CIBERSORTx algorithm. (**B**) The immune score, stromal score, estimate score, and tumor purity in high-risk and low-risk groups assessed by the ESTIMATE algorithm. (**C**) The content of 27 immune cells in diverse risk subclasses assessed by the ssGSEA algorithm. (**D**) Correlation analysis of model genes, Gleason score, risk score and immune cells. * *p* < 0.05, ** *p* < 0.01, *** *p* < 0.001, ns represents no significant difference.

**Figure 7 jcm-12-02243-f007:**
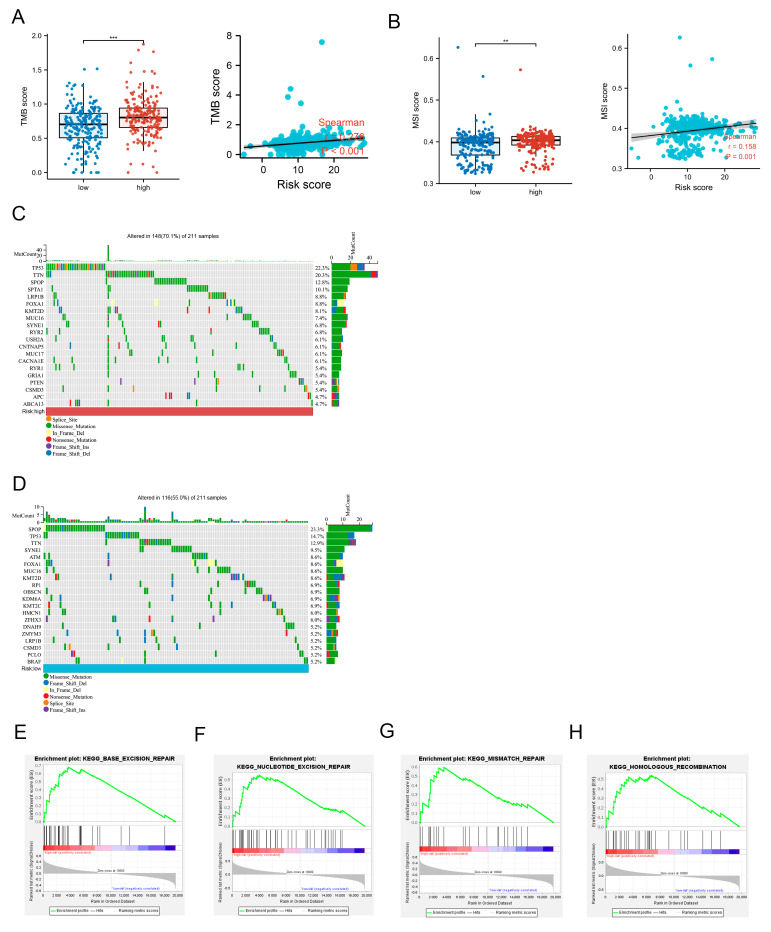
Mutational landscape and single sample GSEA analysis in high-risk and low-risk groups of prostate cancer. (**A**) TMB scores difference between high-risk and low-risk groups, and correlation with risk score. (**B**) MSI scores difference between high-risk and low-risk groups, and correlation with risk score. (**C**) The waterfall plot of top 20 most mutated genes in the high-risk group. (**D**) The waterfall plot of top 20 most mutated genes in the low-risk group. (**E**–**H**) The repair of DNA damage mechanisms of GSEA analysis in the high-risk group. ** *p* < 0.01, *** *p* < 0.001.

**Figure 8 jcm-12-02243-f008:**
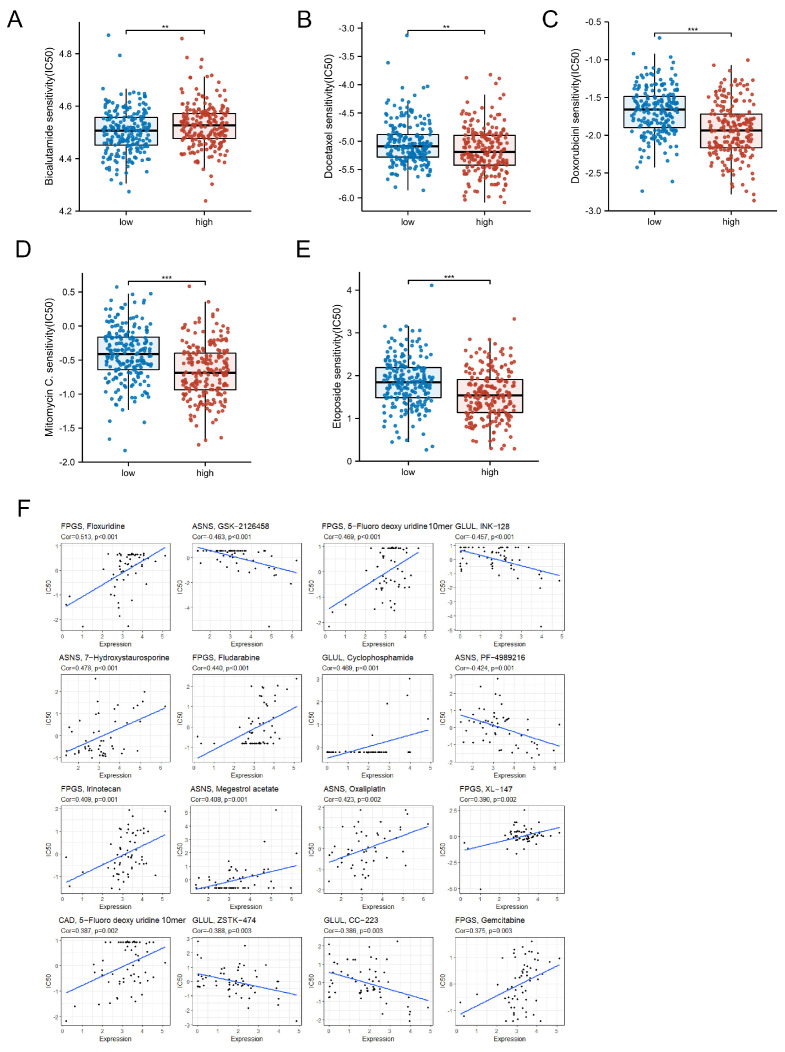
Drug sensitivity analysis in the different risk groups. (**A**–**E**) The IC50 of bicalutamide, docetaxel, doxorubicin, etoposide and mitomycin C difference between high-risk and low risk groups. (**F**) Anticancer sensitivity drug analysis of 5 genes. ** *p* < 0.01, *** *p* < 0.001.

**Figure 9 jcm-12-02243-f009:**
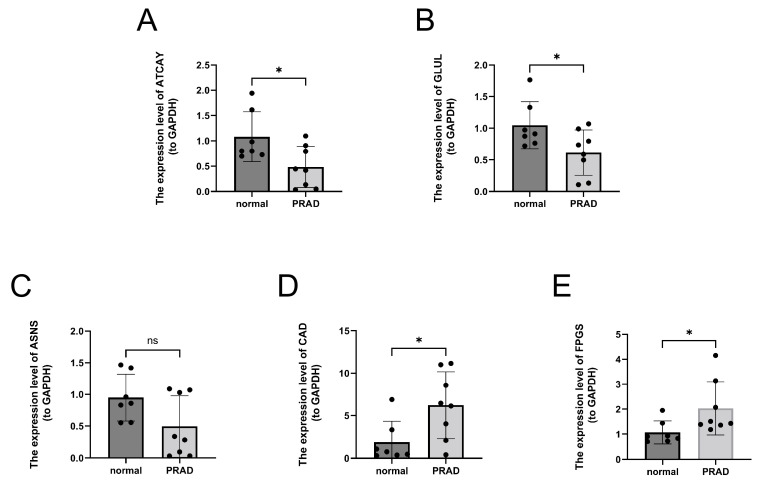
Experimental validation of five key genes’ expression levels in PCa clinical samples. (**A**) The mRNA expression level of ATCAY in mRNA expression levels of prognostic genes in PCa clinical samples. (**B**) The mRNA expression level of GLUL in mRNA expression levels of prognostic genes in PCa clinical samples. (**C**) The mRNA expression level of ASNS in mRNA expression levels of prognostic genes in PCa clinical samples. (**D**) The mRNA expression level of CAD in mRNA expression levels of prognostic genes in PCa clinical samples. (**E**) The mRNA expression level of FPGS in mRNA expression levels of prognostic genes in PCa clinical samples. * *p* < 0.05, ns represents no significant difference.

**Figure 10 jcm-12-02243-f010:**
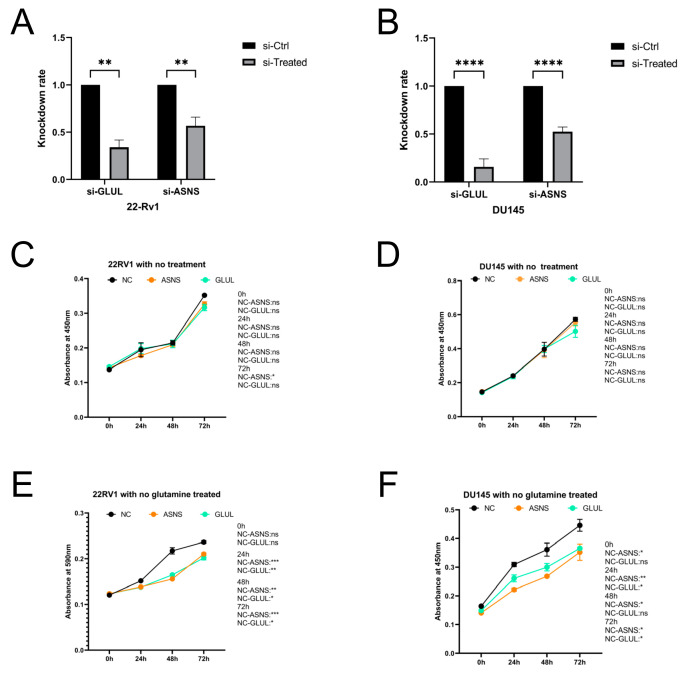
Cell viability of 22Rv1 and DU145 cells after siRNA knockdown of GLUL or ASNS. (**A**,**B**) Knockdown efficiency of siRNA-GLUL and siRNA-ASNS in DU145 and 22Rv1 cell lines by qPCR. (**C**,**E**) The CCK8 proliferation assay of the no treatment groups. (**D**,**F**) The CCK8 proliferation assay of without-glutamine-treated groups. * *p* < 0.05, ** *p* < 0.01, *** *p* < 0.001, **** *p* < 0.0001, ns represents no significant difference.

**Table 1 jcm-12-02243-t001:** Relevant clinical information of 422 PCa patients.

Features	Grade	Detailed Information
Age	>65	119 (28.2%)
	≤65	303 (71.8%)
Gleason score	6	39 (9.2%)
	7	203 (48.1%)
	8	57 (13.5%)
	9	120 (28.5%)
	10	3 (0.7%)
Clinical M stage	M0	397 (94.1%)
	M1	2 (0.5%)
	NA	23 (5.4%)
Clinical N stage	N0	302 (71.6%)
	N1	66 (15.6%)
	NA	54 (12.8%)
Clinical T stage	T2a	7 (1.7%)
	T2b	9 (2.1%)
	T2c	137 (32.5%)
	T3a	135 (32.0%)
	T3b	121 (28.6%)
	T4	8 (1.9%)
	NA	5 (1.2%)
Biochemical recurrence	Yes	55 (13.0%)
	No	367 (87.0%)

**Table 2 jcm-12-02243-t002:** The detailed information of five glutamine-related genes for the prediction model.

Gene ID	Lasso_Coef	HR (95% CI)	*p* Value
ATCAY	−0.204340092	0.510 (0.321–0.813)	0.005
GLUL	−0.130197387	0.635 (0.453–0.890)	0.008
ASNS	0.275033455	2.191 (1.502–3.195)	<0.001
CAD	0.63782178	2.617 (1.569–4.365)	<0.001
FPGS	0.679691817	2.896 (1.724–4.864)	<0.001

## Data Availability

Publicly available datasets were analyzed in this study: The Cancer Genome Atlas (https://portal.gdc.cancer.gov/, accessed on 1 April 2022) and UCSC database (https://xenabrowser.net/datapages/, accessed on 1 April 2022). The GSE70769 dataset was downloaded from Gene Expression Omnibus (GEO: https://www.ncbi.nlm.nih.gov/geo/, accessed on 17 May 2022). The original contributions presented in the study are included in the article/Appendix A. The rest of the data used and analyzed during the current study are available from the corresponding author on reasonable request.

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
