# Peer review of "A Five Glutamine-Associated Signature Predicts Prognosis of Prostate Cancer and Links Glutamine Metabolism with Tumor Microenvironment"

_jcm, 2023, doi:10.3390/jcm12062243_

Round 1
Reviewer 1 Report (Previous Reviewer 1)
Author studied in Silico and in vitro experment intensively. No further request.
Reviewer 2 Report (Previous Reviewer 2)
The authors made corrections and clarifications according to my suggestions.
This manuscript is a resubmission of an earlier submission. The following is a list of the peer review reports and author responses from that submission.
Round 1
Reviewer 1 Report
The author provided five glutamine-associated signatures that predict the prognosis of prostate cancer based on the in silico analysis.
Major
1. The study is in silico analysis without clinical data from the institute. This is not the original article but rather a review article.
2. Although the author identified 5 glutamate-related genes, the prognostic value of the mutated gene is very weak as shown in Figure 6, which makes this paper weak and nonsignificant.
3. Again, as shown in Figure 5, the predictive value of the risk score is very weak compared to that of the Gleason Score and pathological T stage.
Overall, this is just a database analysis, and not an original article.
Reviewer 2 Report
The manuscript entitled “A Five Glutamine-Associated Signature Predicts Prognosis of Prostate Cancer and Links Glutamine Metabolism with Tumor Micro-Environment” describes the bioinformatical approach used for constructing a prediction model for biochemical recurrence of PCa based on the expression of genes involved in glutamine metabolism. Furthermore, the authors use in silico approach to analyze the immunological aspects of PCa and the predicted sensitivity to chemotherapeutics in risk groups stratified according to the same prediction model. Therefore, the main contribution of this manuscript relies on the potential improvements in the accuracy and sensitivity of the prediction models for BCRFS.
The manuscript is well structured, the applied methodology is adequate for this type of study and the main results are generally clearly presented. Still, some clarifications are needed and some issues need to be resolved:
- Substantial English editing is needed, since it is hard to understand some parts of the manuscript. Also, grammar errors and missing spaces are present throughout the manuscript.
- The statements presented in abstract can be misleading, since the results on immunological microenvironment and on the sensitivity to chemotherapy are based on prediction, not on experimental data, which should be clearly stated.
- The “outstanding efficiency” of the model is exaggerated, since the AUC values for ROC curves around 0.6-0.7 suggests relatively poor discrimination ability.
- The Introduction section is focused on the scientific background, but it would benefit from the elaboration of the main concept and the basic hypotheses in relation to the study design and the applied methodology. The last section of the Introduction states the results, not the aims or the basic study design.
- Tregs should be clearly stated as regulatory T cells.
- The rationale for the usage of KEGG and the pathway analysis is not clear, since these genes are preselected based on their involvement in glutamine metabolism. It is not clear what added value this analysis offers, as well as the construction of PPI network. Also, these results were not elaborated in the Discussion section.
- The methodology used for qRT-PCR needs to be described in more detail. The concentrations of primers are not stated, while the reaction mixture composition and the temperature regime for reverse transcription are not presented and there is an obvious error in the presentation of PCR reaction conditions. Namely, each cycle includes two incubation steps (at 96 and 60 degrees), while the other three presented in line 188 on page 4 correspond to melting curve acquisition. In order to make the manuscript more concise and to avoid repetition, primer sequence can be presented in a Table, maybe as a part of supplementary data.
- There is an error in the legend of Figure 1 (line 251, page 7). GSEA should be stated instead of GEO70769.
- The second paragraph on page 8 is redundant, since the direction of changes in gene expression it is obvious from the coefficients presented in Table 2.
- From the results presented in Table 5, it is evident that the HR for the prediction model-generated risk score is relatively low (close to 1), especially in multivariate analysis. This needs to be discussed, taking into account the high HR of Gleason score.
- A discussion would benefit from further elaboration on the results of the study, since some explanations of presented findings are lacking. Furthermore, the limitations should be highlighted.